# Serum vitamin E levels and chronic inflammatory skin diseases: A systematic review and meta-analysis

**Xiaofang Liu[1,☯], Guang Yang[1,☯], Mengxin Luo[2], Qi Lan[2], Xiaoxia Shi[2], Haoyuan Deng[1], Ningning Wang[1], Xuezhu Xu[3]\*, Cong Zhang[1]\***

**1** Department of Food Nutrition and Safety, Dalian Medical University, Dalian, China, **2** Department of Preventive Medicine, Dalian Medical University, Dalian, China, **3** Department of Dermatology, The Second Hospital of Dalian Medical University, Dalian, China

☯ These authors contributed equally to this work.
\* xuxzdl@163.com (XX); congzhang1203@hotmail.com (CZ)

## Abstract

### Background

Vitamin E has long been linked to skin health, including all of its possible functions in cosmetic products, to its roles in membrane integrity and even the aging process. However, reports on the relationship between serum vitamin E levels and the risk of chronic inflammatory skin diseases have been inconsistent. We performed a systematic review and meta-analysis to evaluate the association between serum vitamin E levels and chronic inflammatory skin diseases.

### Methods

We searched the PubMed, Web of Science and Scopus databases, with no time limit up to 30.06.2021. Studies examining serum vitamin E levels in patients with chronic inflammatory skin diseases were selected.

### Results

Twenty articles met the inclusion criteria. Compared with controls, a lower vitamin E level was found in patients with vitiligo (SMD: -0.70, 95% CI: -1.21 to -0.19), psoriasis (SMD: -2.73, 95% CI: -3.57 to -1.18), atopic dermatitis (SMD: -1.08, 95% CI: -1.80 to -0.36) and acne (SMD: -0.67, 95% CI: -1.05 to -0.30).

### Conclusions

Our meta-analysis showed that serum vitamin E levels were lower in patients suffering from vitiligo, psoriasis, atopic dermatitis and acne. This study highlights the need to evaluate vitamin E status to improve its level in patients with skin diseases.

**Data Availability Statement:** All relevant data are within the paper and its Supporting Information files.

**Funding:** This work was supported by the Basic Research Projects of Colleges and Universities of

Liaoning Province, China (LQ2017037). Recipient: Cong Zhang. The funders had no role in study design, data collection and analysis, decision to publish, or preparation of the manuscript.

## Introduction

Skin disease is a highly prevalent disease and is estimated to affect 30% -70% of individuals of all cultures and ages, causing a substantial burden worldwide. Skin disease has become a global public health problem [1–3]. Chronic inflammatory skin disease is an umbrella term grouping heterogeneous entities characterized by chronic immunologically driven skin inflammation, and encompasses many common disorders, including atopic dermatitis, psoriasis, lichenoid reactions, autoimmune bullous diseases and some granulomatous reactions [4,5]. These diseases are difficult to cure and often accompanied by long-term symptoms such as itching, pain and skin damage, which can diminish a patient's quality and even length of life [6–8]. Chronic inflammatory skin disease also be associated and often coexisting with obesity, cardiovascular disease, diabetes mellitus and many other immune-related clinical conditions [9–11]. The pathophysiology of chronic inflammatory skin diseases is unclear, but it is presumed to be an ensemble of genetic and environmental aspects leading to an impaired immunological activation in patients of chronic inflammatory skin diseases.

Vitamin E is an essential and beneficial nutrient in various aspects of health that is attracting growing attention in skin care. Vitamin E can be divided into two groups, tocopherols and tocotrienols, with four isomers (alpha, beta, gamma and delta). Compared to the exogenous lipophilic vitamins that exist in the body, vitamin E was shown to be more evenly distributed in the whole body, especially in the plasma [12,13]. Aberrant changes in serum vitamins have been reported in chronic inflammatory skin diseases [14,15]. To date, studies on the level of serum vitamin E and skin diseases have mainly focused on chronic inflammatory skin diseases such as vitiligo, psoriasis, atopic dermatitis and acne [16–35]. Most of these studies generally support an inverse association between serum levels of vitamin E and the risk of skin diseases. However, the result is still controversial because other studies show that there is no relationship between serum levels of vitamin E and the risk of skin diseases. There is lack of meta-analysis on the relationship between serum levels of vitamin E and chronic inflammatory skin diseases.

Based on the results of previous studies, we speculated that the increased risk of chronic inflammatory skin diseases in some populations may partly be explained by their poor serum vitamin E status. As a first step in addressing this issue, we performed a systematic review and meta-analysis to determine the relationship between serum levels of vitamin E and chronic inflammatory skin diseases including vitiligo, psoriasis, atopic dermatitis and acne.

## Materials and methods

### Literature search

This meta-analysis was registered on Prospero (CRD42020207143). All literatures investigating the association of serum vitamin E levels and skin diseases were conducted in according to the Preferred Reporting Items for Systematic Reviews and Meta-Analyses (PRISMA) [36]. The studies were screened and selected independently by two authors (Q.L. and MX.L.). The quality assessment of articles was also performed by the two authors. Any disagreement was resolved through discussion with a third author (C.Z.).

The literature search on relationship between serum vitamin E levels and chronic inflammatory skin diseases was performed in PubMed, Scopus, and Web of Science databases and considered articles published until 30.06.2021. We included English-language studies of all designs. The literature search strategy was showed in S1 Table.

### Eligibility criteria and study selection

Full-text articles were eligible if they met the following inclusion criteria: studies reporting relevant topics on comparing serum vitamin E levels in patients with chronic inflammatory skin diseases (vitiligo, atopic dermatitis, psoriasis or acne) and the control groups of healthy individuals without skin diseases. The review articles, case reports, animal studies and in vitro experiments were excluded from analysis.

### Data extraction

Data were extracted from each study by two authors using a standardized data-collection protocol. In each study, the following information was extracted in tabular form: first author, publication year, country, number of cases and controls, skin diseases, patient gender and age, serum vitamin E determination method, and levels of serum vitamin E.

### Data quality assessments

The Newcastle–Ottawa Scale was used to assess the quality of included case control studies [37]. Using this scale, every individual study is judged on eight items, categorized into three groups: selection of study groups; comparability of groups; and ascertainment of exposure/outcome. Stars are given for each quality item and the highest quality studies are given up to nine stars. A study is considered of high-quality if there are 3 or 4 stars in selection domain AND 1 or 2 stars in comparability domain AND 2 or 3 stars in exposure/outcome domain [37]. Studies with $\geq$ 6 stars, 4–6 stars were considered high and moderate quality, respectively. Studies with $\leq$ 4 stars were considered low quality and were excluded [38].

### Statistical analysis

All analyses were performed using Stata software (Stata Statistical Software, Release 13; Stata-Corp LP, College Station, TX, USA). The standard mean differences (SMD) and corresponding 95% confidence intervals (CI) were pooled to evaluate the association between serum vitamin E levels and different chronic inflammatory skin diseases, as well as the association between serum vitamin E levels and disease severity. Heterogeneity test was performed by $I^2$ statistic [39]. A random effects model was used because of the high heterogeneity ($I^2 > 50\%$). Publication bias was evaluated by using Egger's test.

To explore heterogeneity, subgroup and meta-regression were performed to determine the effects of the study quality (< 6 vs. $\geq$6 stars), region (Europe vs. Asia), study size (n < 100 vs. n $\geq$ 100), age (children vs. other), gender (female > male vs. female < male vs. female = male) and whether gender or age-matched controls (yes vs. no) on the levels of serum vitamin E levels of patients with chronic inflammatory skin diseases and controls. Sensitivity analysis was performed to try to explain the influence of individual studies on the overall effect size.

## Results

### Literature search

The search screened 892 studies: 192 from PubMed, 58 from Scopus, and 642 from Web of Science. After carefully selection and exclusion, 20 case-control studies involving a total of 1172 patients were included into this review (Fig 1) [16–35].

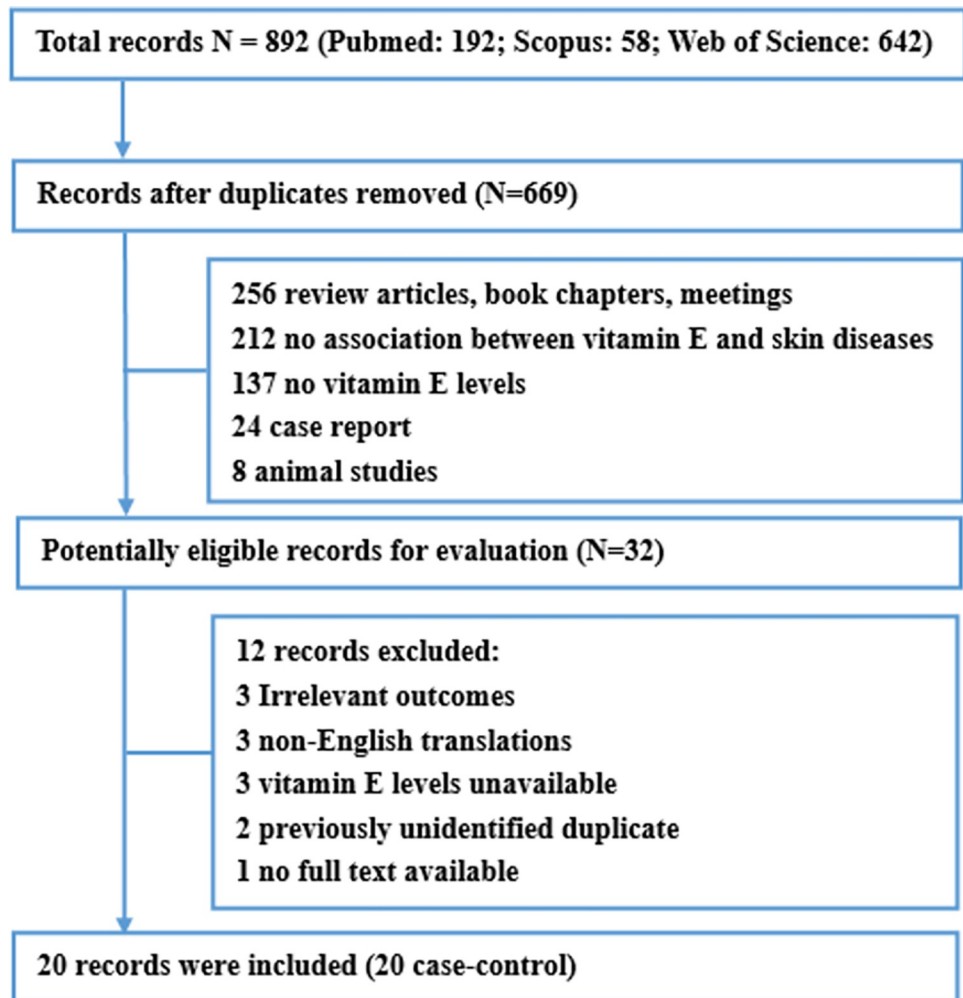

**Fig 1. Flow diagram of systematic literature search.**

## Study characteristics

The main relevant information of the 20 case-control studies included in our review was summarized in Table 1. Studies were mainly focus on chronic inflammatory skin diseases, including vitiligo, psoriasis, atopic dermatitis and acne. Six studies were from India; four from Turkey, two each from Italy and Poland; and one each from South Korea, German, Portugal, Jordan, Nepal and Tunisia. Five studies included only children or teenagers, eight included only adults, six included children and adults and one without age description. Eleven studies adopted the measurement of high-performance liquid chromatography (HPLC), three with fluorescence spectrophotometry (FS), two with gas chromatography mass spectrometry (GC-MS), one with rapid determination of vitamin E assay, one with Baker and Frank's protocol, and one with α-α dipyridyl antioxidant assay. Thirteen studies elaborated that alpha tocopherol was estimated in their investigations, seven studies did not describe the subunit of measured vitamin E. Serum vitamin E levels of different skin diseases were showed in S2 Table.

In quality assessment, scores of all included case-control studies are showed in S3 Table. Most studies were high quality as assessed by the NOS. For selection of study groups, 11 of the 20 studies scored 3 or 4 stars. For comparability, all of the 20 studies scored 1 or 2 stars. For assessment of outcome, all of the 20 studies provided ascertainment of exposure, and had same

**Table 1. Main characteristics of studies included in the review.**

| Author (year) | Country of origin | Children*, adults or both | Gender | n, cases | n, total | Type of control | Method of vitamin E determination | Subsets of vitamin E | Skin disease | Method of Skin diseases determination | Matching or adjustment factors |
|---|---|---|---|---|---|---|---|---|---|---|---|
| Oh et al., 2010 [17] | Korea | Young children | Patients: 54.7% boys, 45.3% girls; Control: 50.8% boys, 49.2% girls | 180 | 422 | Non- atopic Dermatitis | HPLC | alpha tocopherol | Atopic Dermatitis | SCORAD | Gender, Age |
| Sivaranjani et al., 2013 [32] | India | An average age of 35 years (10–60 years) | NA | 25 | 50 | Healthy controls of same age group | Spectrophotometry | alpha tocopherol | | NA | Age |
| Daniluk et al., 2019 [16] | Poland | Children aged from 1 to 15 years | Patients: 41.4% boys, 58.6% girls; Control: 63.6% boys, 36.4% girls | 29 | 51 | Healthy controls with negative history of allergy | HPLC | alpha tocopherol | | SCORAD | Age |
| Hozyasz et al., 2004 [25] | Poland | Children with DA (age: 1–9 years). | NA | 25 | 43 | Healthy controls | HPLC | alpha tocopherol | | Clinically diagnosed | Age |
| Ines et al.,2006 [18] | Tunisia | 18–66 years old | Patients: 38.9% male, 61.1% female; Control: 37.5% male, 62.5% female | 36 | 76 | Healthy controls | HPLC | NA | Vitiligo | Clinically diagnosed | Age |
| Khan et al., 2009 [19] | India | 22–41 years old aldult | Patients: 23.3% male, 76.7% female; Control: matched | 30 | 60 | Healthy controls | Fluorometery | alpha tocopherol | | Clinically diagnosed | Gender, Age |
| Agrawal et al., 2014 [22] | Nepal | <15 years old | Patients: 48% male, 52% female; Control: 45% male, 55% female | 80 | 160 | Healthy controls | Spectrophotometric assay | alpha tocopherol | | Clinically diagnosed | Age |
| Jain et al., 2008 [26] | India | 11–20 years old | Patients: 50% male, 50% female; Control: 50% male, 50% female | 40 | 80 | Healthy controls | Rapid determination of vitamin E | NA | | Clinically diagnosed | Gender, age |
| Dell'Anna et al., 2001 [23] | Italy | 18–53 years old | Patients: 50% male, 50% female; Control: 50% male, 50% female | 40 | 80 | Healthy controls | Gas Chromatography Mass Spectrometry (GC-MS) | NA | | Clinically diagnosed | Gender, age |
| Agrawal et al., 2004 [22] | India | 5–45 years old | Patients: 37.5% male, 62.5% female; Control: 37.5% male, 62.5% female | 63 | 123 | Healthy controls | Spectrophotometry | NA | | Clinically diagnosed | Age |
| Picardo et al., 1994 [29] | Italy | 19–45 years old | Patients: 40.3% male, 59.7% female; Control: 40.3% male, 59.7% female | 62 | 122 | Healthy controls | GC-MS | alpha tocopherol | | Clinically diagnosed | Age |

(*Continued*)

**Table 1.** (Continued)

| Author (year) | Country of origin | Children*, adults or both | Gender | n, cases | n, total | Type of control | Method of vitamin E determination | Subsets of vitamin E | Skin disease | Method of Skin diseases determination | Matching or adjustment factors |
|---|---|---|---|---|---|---|---|---|---|---|---|
| Kökçam et al., 1999 [28] | Turkey | 9 to 76 years old | Patients: 58.8% male, 41.2% female; Control: matched | 34 | 68 | Healthy controls | HPLC | alpha tocopherol | Psoriasis | Clinically diagnosed | Gender, age |
| Pereira et al., 2004 [21] | Portugal | Patients:47.4 ± 13.3 years old;Control:45.9 ± 12.2 years old | Patients: 57% male, 43% female; Control: 55% male, 45% female | 70 | 110 | Healthy controls | HPLC | alpha tocopherol | | Clinically diagnosed | Gender, age, BMI |
| Jain, et al., 1988 [27] | India | Both sexes from various age groups | Both sexes | 20 | 40 | Healthy controls | By method of Baker and Frank | NA | | Clinically diagnosed | Gender, age |
| Demir et al., 2013 [24] | Turkey | Patients:39.23 ± 15.32 years old; control: 38.46 ± 10.32 years old | Patients: 32.3% male, 67.7% female; Control: 35.1% male, 64.9% female | 31 | 68 | Healthy controls | HPLC | alpha tocopherol | | Biopsy-proven diagnosed | Gender, age |
| Pujari et al., 2014 [30] | India | Aged 20–60 years | NA | 90 | 180 | Healthy controls | α-α dipyridyl antioxidant assay | NA | | Clinical diagnosed | Gender, age |
| Severin et al., 1999 [31] | Germany | NA | NA | 33 | 69 | Healthy controls | HPLC | alpha tocopherol | | Clinical diagnosed | NA |
| El-akawi et al., 2006 [35] | Jordan | Patients: 21.0 ± 5.4 years old; controls: 21.3 ± 5.3 years old | NA | 100 | 200 | Healthy controls | HPLC | NA | Acne | GAGS | Age |
| Ozuguz et al., 2013 [33] | Turkey | Patients: 28.54±8.30 years old | Patients: 35.1% male, 64.9% female; Control: 21.4% male, 78.6% female | 94 | 150 | Healthy controls | HPLC | alpha tocopherol | | Clinical diagnosed | Gender, age |
| Tunçez Akyürek | Turkey | Patients:18.67 ± 3.36 years old; controls: 19.7 ± 2.49 years old | NA | 90 | 120 | Healthy controls | HPLC | alpha tocopherol | | GAGS | NA |

*Children defined as <18 years of age.

HPLC: High-performance liquid chromatography.

SCORAD: Scoring Atopic Dermatitis index.

GAGS: Global Acne Grading System.

method of ascertainment for cases and controls, but only one study provided non-response rates or described non-respondents.

## Serum vitamin E levels and vitiligo

Levels of serum vitamin E in patients with vitiligo were reported in 7 studies, with 351 cases and 350 controls in total (p < 0.001, $I^2$ = 90.4%). Compared with the control group, vitiligo patients had significant lower level of serum vitamin E (SMD: -0.70, 95% CI: -1.21 to -0.19) (Fig 2). No publication bias was detected by Egger's test (p = 0.248).

## Serum vitamin E levels and psoriasis

Six studies investigated the change of serum vitamin E levels in psoriasis, with a total of 278 cases and 257 controls (p < 0.001, $I^2$ = 97.6%). Compared with the control group, psoriasis

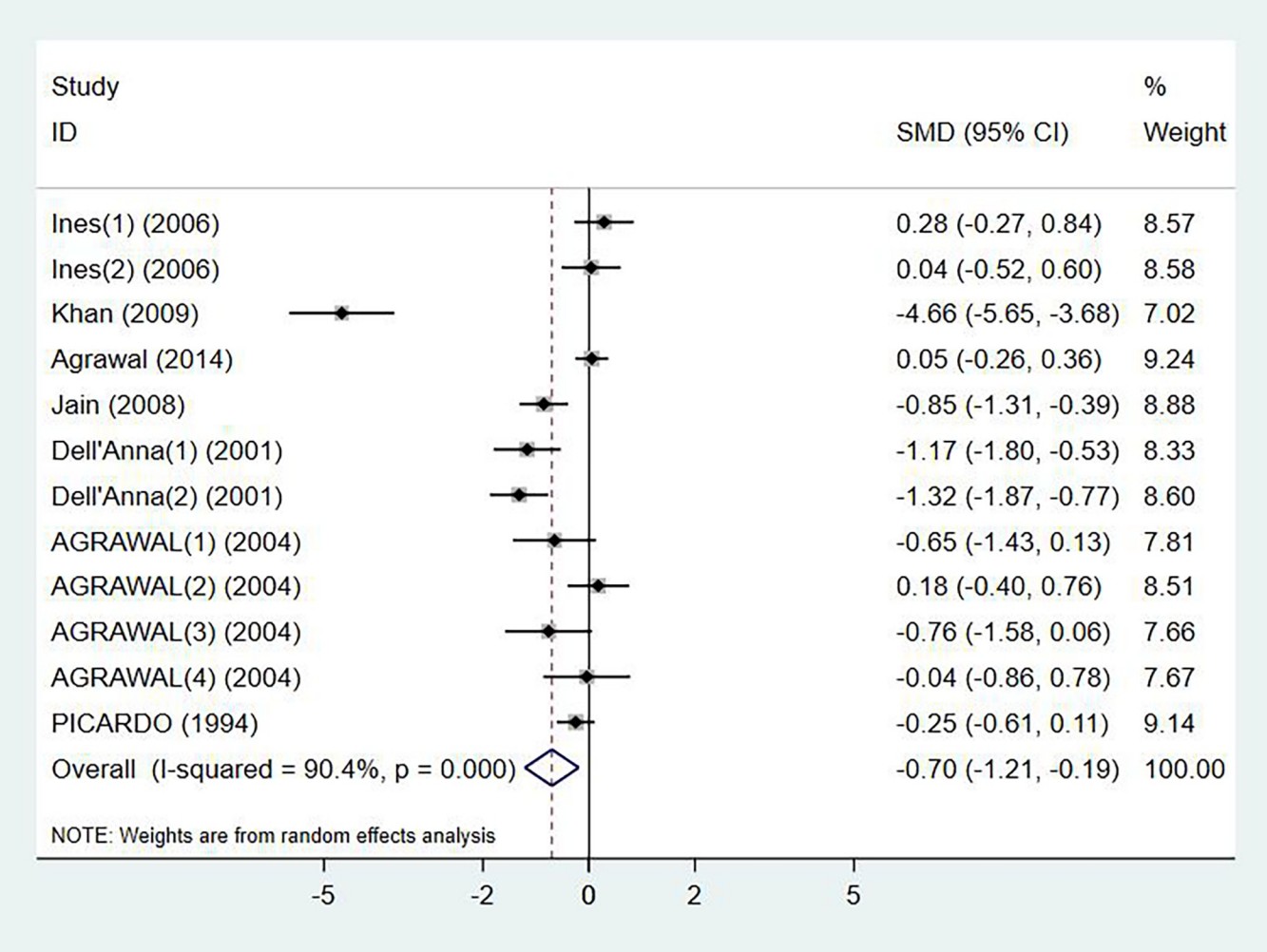

**Fig 2. Forest plot in the meta-analysis of vitamin E levels and vitiligo.**

patients had significant lower level of serum vitamin E (SMD: -2.37, 95% CI: -3.57 to -1.18) (Fig 3). No publication bias was detected by Egger's test (p = 0.058).

## Serum vitamin E levels and atopic dermatitis

The serum vitamin E Levels in patients with atopic dermatitis were observed in 4 studies, with 259 cases and 307 controls in total (p < 0.001, $I^2$ = 91.1%). Compared with the control group, atopic dermatitis patients had significant lower level of serum vitamin E (SMD: -1.08, 95% CI: -1.80 to -0.36) (Fig 4). No publication bias was detected by Egger's test (p = 0.677).

## Serum vitamin E levels and acne

Levels of serum vitamin E in acne patients were reported in 3 studies, with 284 cases and 186 controls in total (p = 0.240, $I^2$ = 83.9%). Compared with the control group, acne patients had significant lower level of serum vitamin E (SMD: -0.67, 95% CI: -1.05 to -0.30) (Fig 5). No publication bias was detected by Egger's test (p = 0.879).

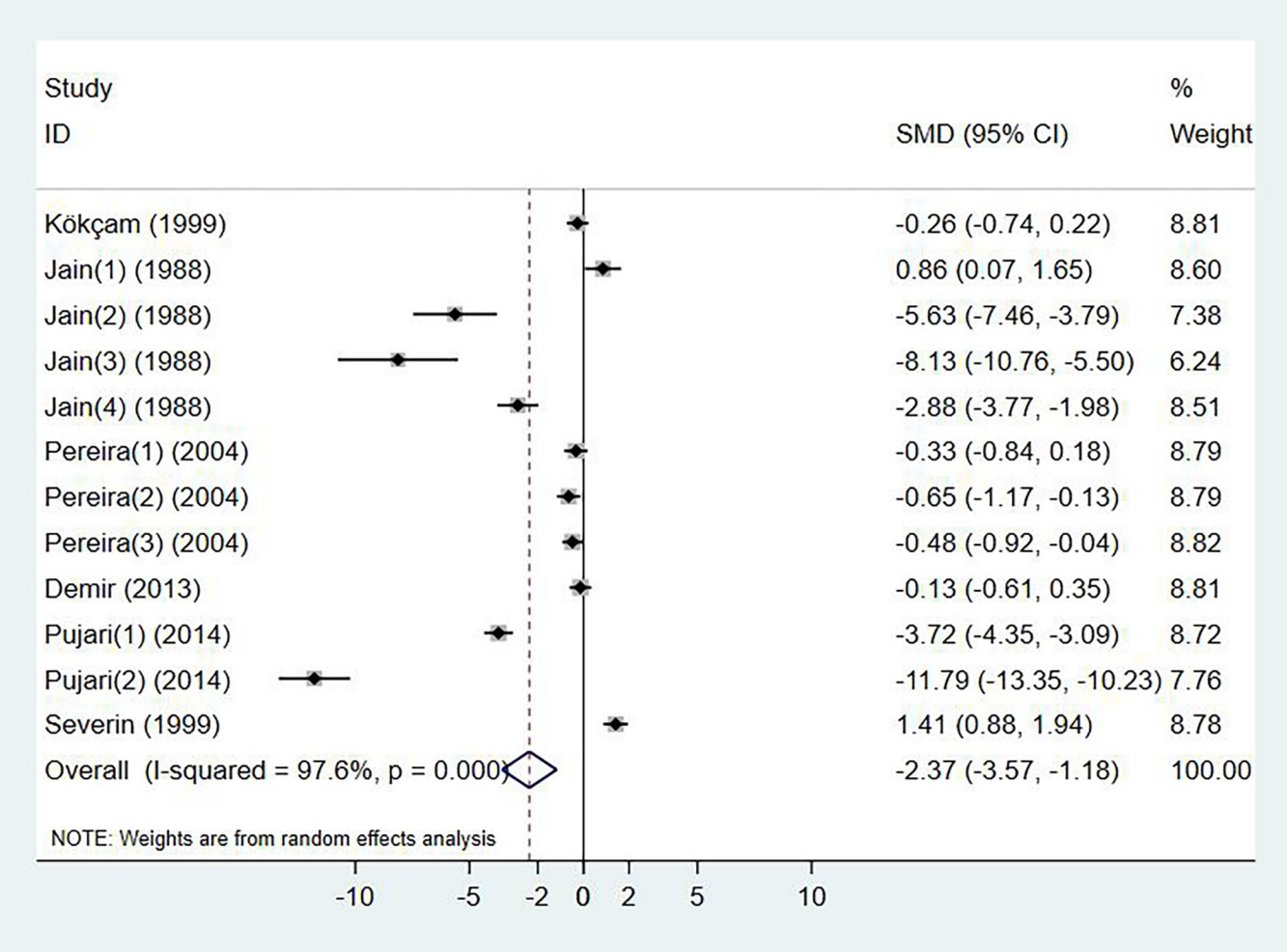

**Fig 3. Forest plot in the meta-analysis of vitamin E levels and psoriasis.**

## Serum vitamin E levels and skin disease severity or stages

As showed in Fig 6, the level of serum vitamin E in active vitiligo group was similar to that observed in the stable vitiligo group, regardless of disease activity (SMD: -0.08, 95% CI: -0.45 to 0.30, n = 3), and no publication bias was detected by Egger's test (p = 0.434). Compare with mild psoriasis patients, severe psoriasis patients had a lower serum vitamin E level without significant (SMD: -8.35, 95% CI: -17.46 to 0.76, n = 3), and no publication bias was detected by Egger's test (p = 0.527). Severe atopic dermatitis patients had a significantly lower level of serum vitamin E compare with mild patients (SMD: -0.93, 95% CI: -1.76 to -0.10, n = 1). The level of serum vitamin E has no significant differences between severe and mild acne patients (SMD:-0.43, 95% CI: -1.32 to 0.45, n = 2).

## Subgroup analysis and Sensitivity analysis

Because of high heterogeneity existed in the meta-analysis (>70%), we conducted subgroup and meta-regression analyses based on study quality, region, study size, age, gender, and whether gender or age-matched. Our result showed no modification by study quality, region,

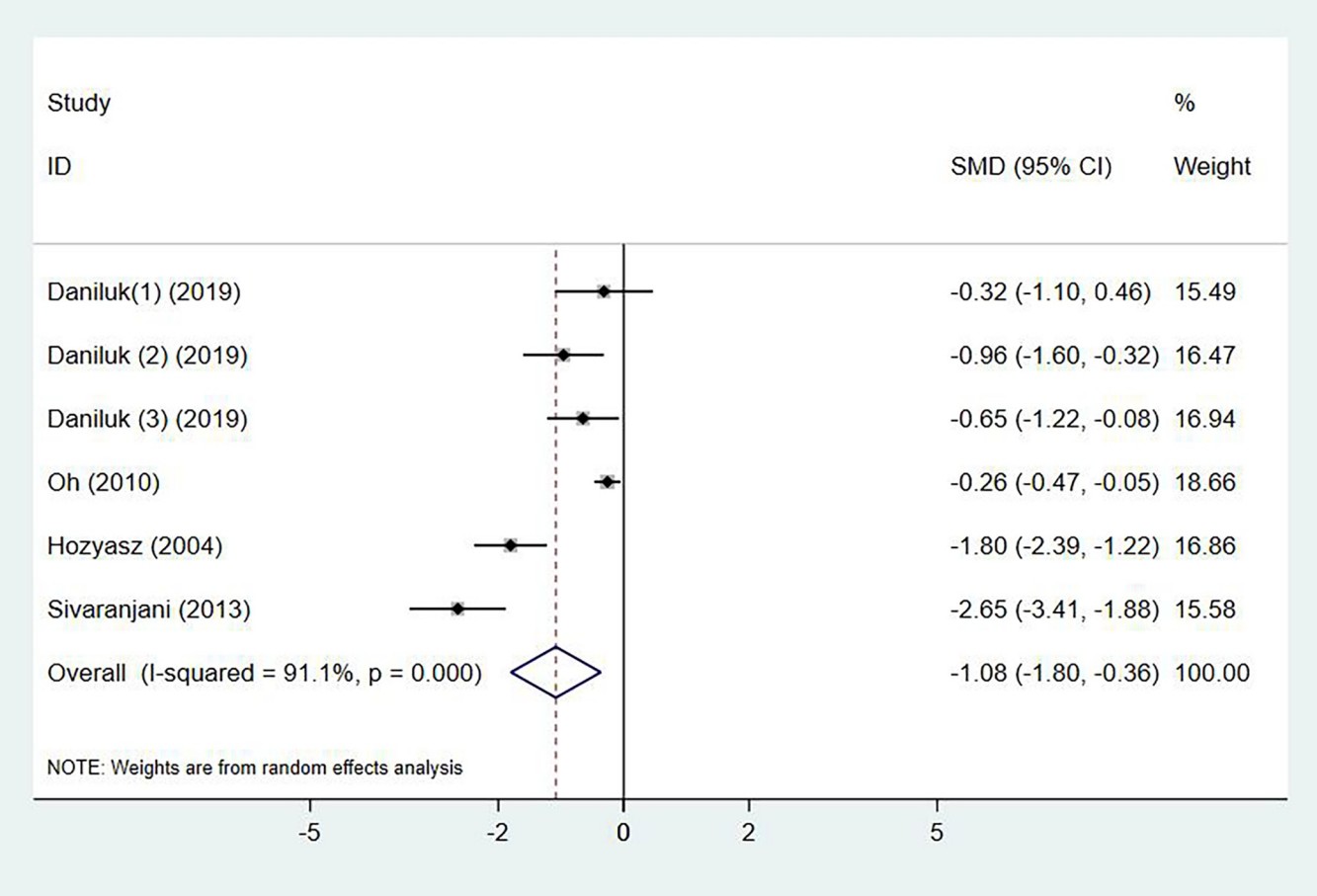

**Fig 4. Forest plot in the meta-analysis of vitamin E levels and atopic dermatitis.**

study size, age, gender, and whether sex or age-matched on the relationship between serum vitamin E levels and skin diseases (S4 Table).

In a sensitivity analysis, a meta-influence plot was used to analyze the influence of individual studies on the overall effect size. As shown in Fig 7, no single study has influence on the above-mentioned pooled effect of the association between serum vitamin E levels and skin diseases, further confirmed the robust of our findings.

## Discussion

As a dietary bioactive compound, vitamin E is very important for skin health. The level of vitamin E and its importance in the pathogenesis of skin disease has been evaluated in multiple investigations. Although most investigations showed a decrease in serum vitamin E levels in patients with skin diseases, some reports showed no significant change in the serum vitamin E status of patients with skin diseases. Previous studies have inspired the idea that the level of serum vitamin E is related to skin diseases. Therefore, we summed up the results of studies on serum levels of vitamin E in patients with skin diseases, and our study was indicating the lower vitamin E levels in patients with chronic inflammatory skin diseases such as vitiligo, psoriasis, atopic dermatitis and acne.

Chronic skin diseases are related to immune disorders that involve the interplay between oxidative stress and the immune system. Recently, oxidative stress and the accumulation of

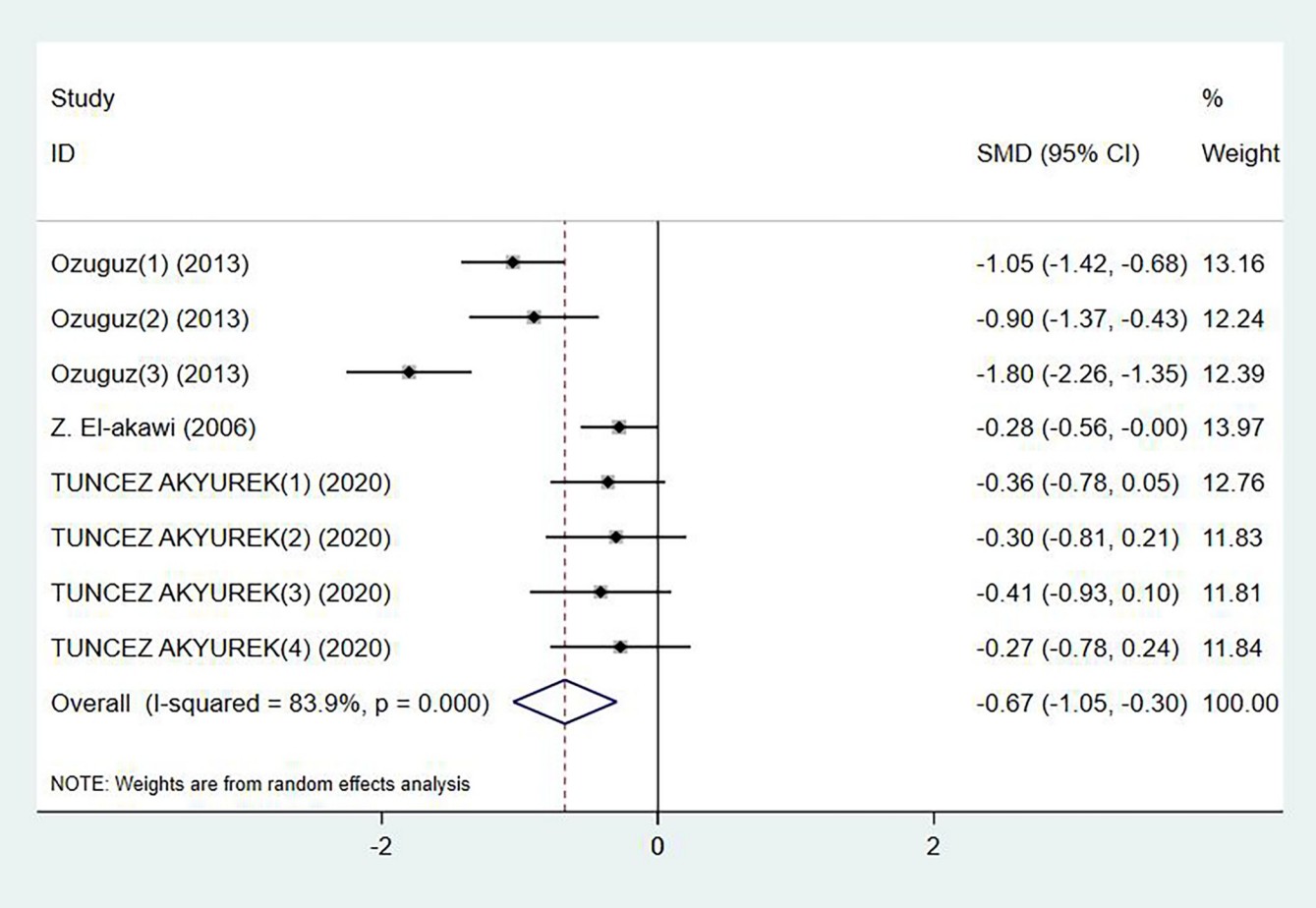

**Fig 5. Forest plot in the meta-analysis of vitamin E levels and acne.**

free radicals in the epidermal layer of affected skin have been shown to be involved in the pathophysiology of vitiligo [40]. In patients with atopic dermatitis, urine 8-hydroxydeoxygua-nosine which is a product of DNA oxidation by free radicals, levels were higher than those in healthy people [41]. Accordingly, the imbalance of Th2 to Th1 cytokines can create alterations in cell-mediated immune responses and can promote IgE-mediated hypersensitivity [42]. As another clinically inflammatory skin disease, psoriasis can occur due to abnormalities in essential fatty acid metabolism, lymphokine secretion, lipid peroxidation and free radical generation [43]. Propionibacterium acnes (P. acnes) produces a variety of chemical factors that induce neutrophil chemotaxis, while neutrophils attempt to attack P. acnes by ROS secretion, which can initiate inflammation in normal tissues, leading to acne skin symptoms [44]. Vitamin E has gained researchers' attention as a potential adjuvant therapy for various skin disorders due to its excellent antioxidant and anti-inflammatory properties. Vitamin E supplementation resulted in the significant improvement of clinical conditions and normalization of oxidative stress markers in patients with psoriasis and vitiligo [45,46]. In our study, there were lower serum vitamin E levels in patients with vitiligo, psoriasis, atopic dermatitis and acne than in controls. We believe this might depend on the antioxidant function as well as the many other important functions by vitamin E, such as its essential role in the function of neutrophils and it also influence lipid metabolism or directly affect T cells in immune response [47]. The exact

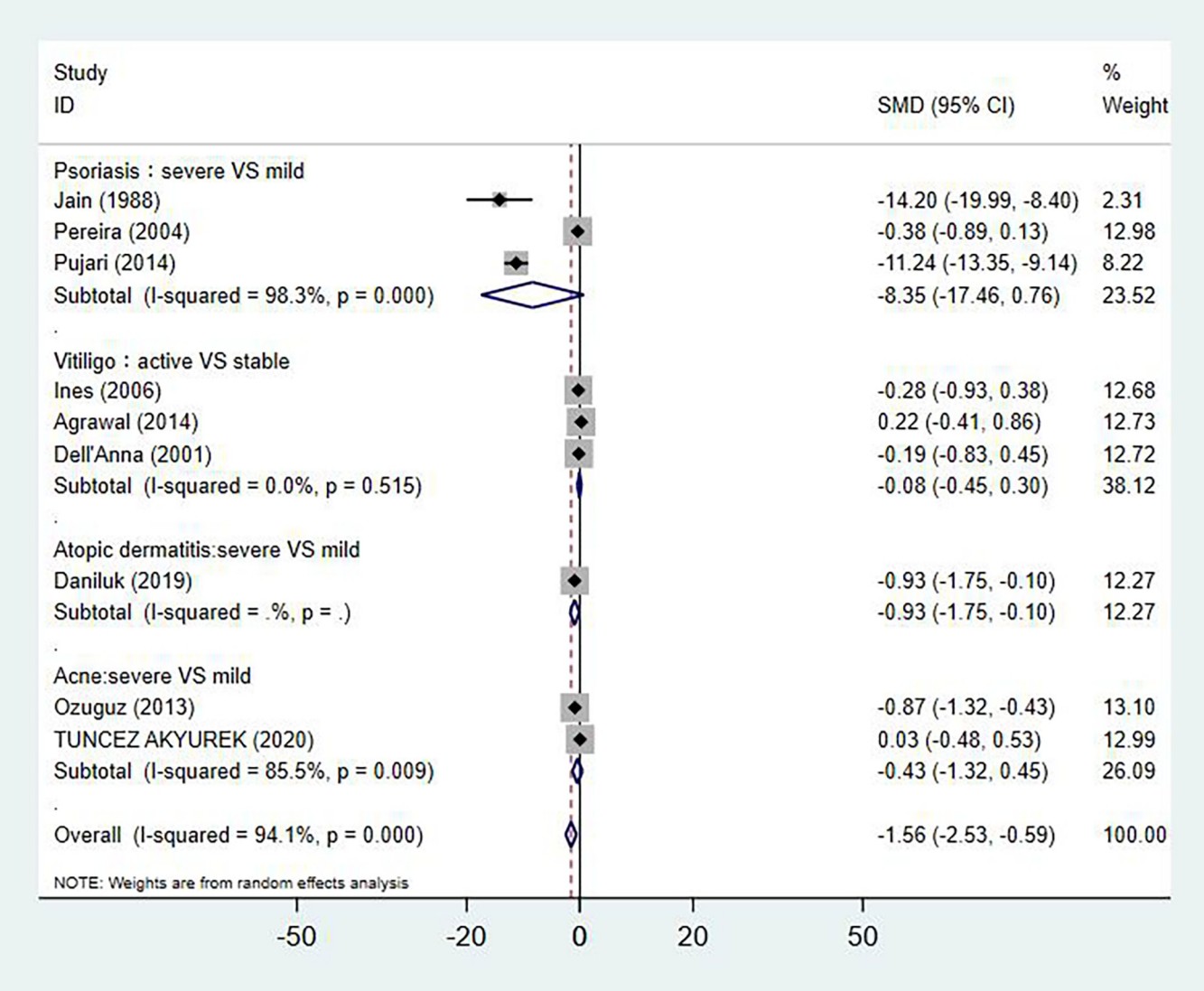

**Fig 6. Forest plot in the meta-analysis of vitamin E levels and skin disease severity.**

mechanisms of serum vitamin E levels and chronic inflammatory skin diseases need further investigation.

Apart from immune disorders/autoimmunity, oxidative stress and genetic predisposition, environmental risk factors including lifestyle behaviors are the main determinants of skin diseases, especially dietary patterns. An increasing body of research indicates that dietary changes may play an important role in chronic skin diseases, especially nutrients that have bioactive functions such as antioxidant or anti-inflammatory activities [48]. It has also been found that dietary habits play an important role in pathogenesis and treatment of chronic skin diseases including psoriasis, vitiligo, acne and atopic dermatitis [49–52]. Deficiency of vitamin E has been shown to cause skin anomalies [53]. Natural sources of vitamin E include nuts, plant-based oils and vegetables. It was found that >60% of adults have vitamin E intakes below the EAR (<12 mg/d) in the United States [54,55]. Vitamin E deficiencies are more frequently found in children, likely because they have limited stores and are growing rapidly [56]. Many

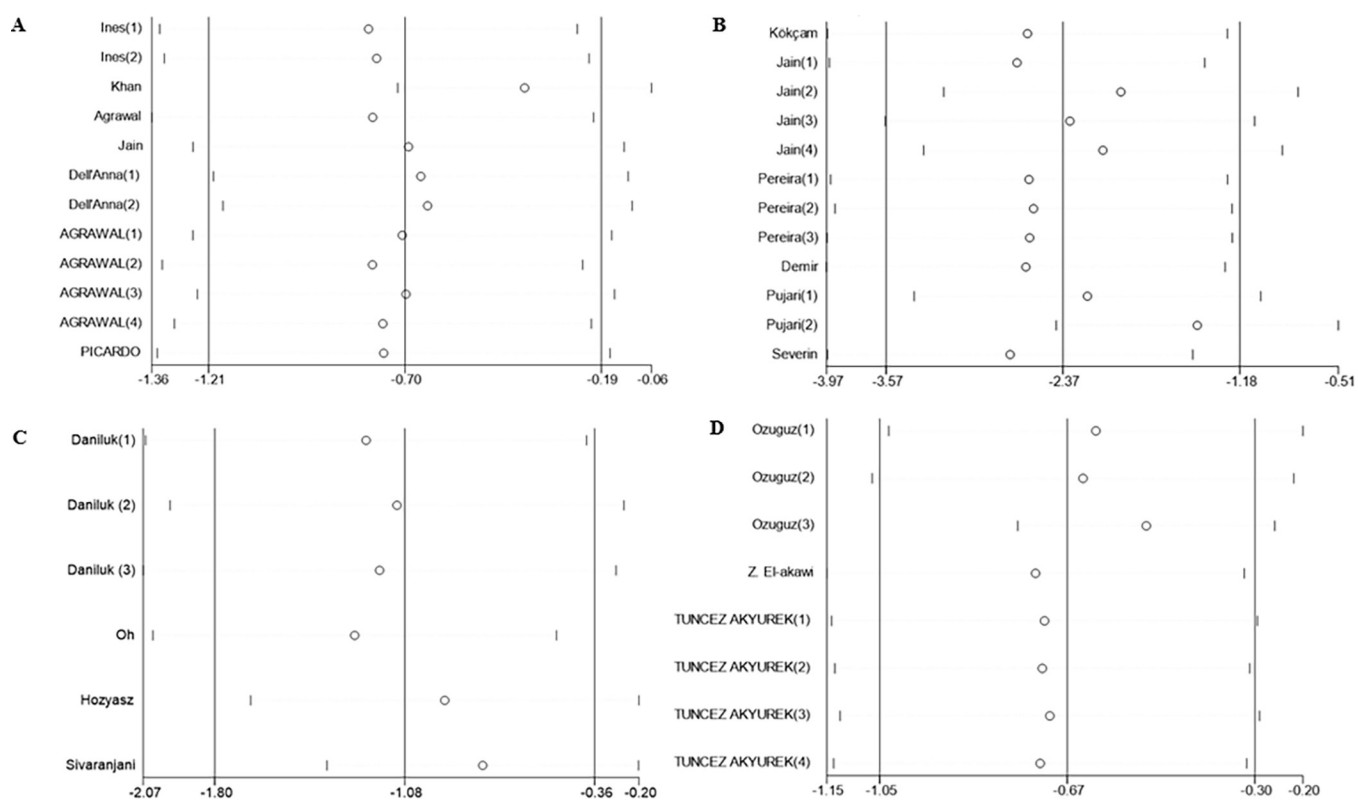

**Fig 7. Sensitivity analysis of studies included in meta-analysis.** A: Vitiligo; B: Psoriasis; C: Atopic dermatitis; D: Acne.

studies of dietary interventions on skin diseases have proven that nutrient supplementation is an auxiliary means for the treatment of skin diseases. Elgoweini et al. founded that vitamin E supplementation prevented lipid peroxidation in the cellular membrane of melanocytes and increased the effectiveness of NB-UVB in patients with stable vitiligo [57]. Goforoushan et al. reported that vitamin E prevents dermal complications of isotretinoin [58]. Oh et al. observed a lower likelihood of atopic dermatitis in children who consumed more dietary vitamin E, which led to higher serum vitamin E levels [52]. Therefore, the importance of maintaining serum vitamin E levels might be suggested for patients diagnosed with skin diseases.

Some clinical studies have been investigated as a mainstay treatment of vitamin E in skin diseases. Two randomized, double-blind, placebo-controlled clinical trials (RCTs) using the same model performed by Javanbakht et al. and Jaffary et al. concluded that alpha tocopherol increased erythrocyte superoxide dismutase activity and reduced the Scoring Atopic Dermatitis index in patients with atopic dermatitis [59,60]. A clinical trial conducted by Tsoureli-Nikita et al. showed that in patients with atopic dermatitis, treatment with oral vitamin E (400 IU/day) for 8 months had observable reductions in skin lesions and pruritus, comparable to treatment with topical corticosteroid and oral antihistamines [61]. One RCT performed by Kharaeva et al. found that supplementation with antioxidants containing vitamin E resulted in the significant improvement of clinical conditions and normalization of oxidative stress markers in patients with severe erythrodermic and arthropathic forms of psoriasis [46]. Improving vitamin E levels in patients with skin diseases might have a positive effect. However, much more research is needed on this subject due to the small amount of currently published studies.

Between-study heterogeneity is common in meta-analyses and it is essential to explore its potential sources. We conducted subgroup and meta-regression analyses on variables including study quality, region, study size, age (children vs. other), gender, and whether gender or age-matched controls to explore the potential sources of heterogeneity. However, these factors were not found to be sources of heterogeneity in our meta-analysis, but other possibilities related to skin diseases, such as variations in lifestyle and dietary practices, cannot be ruled out. Furthermore, a sensitivity analysis was performed to assess the influence of individual studies on the overall effect size. Our findings did not change significantly with the removal of any of the studies, suggesting that the association of interest is minimally impacted by the inclusion of these studies.

Several limitations should be considered for interpretation of the results. First, the number of studies for each skin disease was relatively small. Second, dietary vitamin E can vary significantly by country, religion, geographic location and food culture. Third, the studies was heterogeneous in several aspects, such as method of vitamin E determination and diagnostic criteria for skin diseases. Fourth, it is worth noting that an individual's response to vitamin E varies depending on several factors including health condition, nutritional status, medication effect, life stage, and genetic heterogeneity. The reported associations may be confounded by other nutrients due to the synergistic or antagonistic effects of nutrients, and/or lifestyle factors known to influence the risk of skin diseases. Future large-scale studies are warranted to explore such combined effects.

Additionally, there were no differences in serum vitamin E levels between patients with active vitiligo and patients with stable vitiligo, patients with severe acne and patients with mild acne, or patients with severe psoriasis and patients with mild psoriasis, although there was a significantly decreased serum vitamin E level in patients with severe atopic dermatitis compare with patients with mild atopic dermatitis. This might be because the number of studies on serum vitamin E levels and skin disease severity is different between atopic dermatitis (n = 1) and the others (n>1). The inadequate number of studies indicates the need for the consensus on definitions for low levels of vitamin E and skin disease severity in future studies.

Finally, except for 7 studies that did not report vitamin E subtypes in their measurements, 13 of 20 studies elaborated that alpha tocopherol (α-tocopherol) was estimated in their investigations. Are other tocopherols associate with human chronic inflammatory skin diseases? The α-tocopherol structure is unique and not substitutable with other tocopherols or tocotrienols, because functions attributed to vitamin E were not only antioxidant and anti-inflammation. There are many possible mechanisms like gene expression regelation, microRNA modifications, tocopherol associated proteins, albumin binding and stem cell differentiation, as well as active forms of vitamin E including tocopheryl phosphate and vitamin E metabolites could be involved in vitamin E's benefit [62], which we should pay attention now and explore in future.

## Conclusion

Our review and meta-analysis indicated that lower serum vitamin E levels were associated with several chronic inflammatory skin diseases, such as vitiligo, psoriasis, atopic dermatitis, and acne. This is an important finding in terms of future randomized controlled trials that might reveal a direct cause and effect relationship between vitamin E intake, as well as serum vitamin E levels, and the risk of skin diseases. Furthermore, the potential mechanism associated with the preventive effects of vitamin E should be further elucidated in future studies.

## Supporting information

**S1 Checklist. PRISMA 2020 checklist.**
(DOCX)

**S1 Table. Systematic search strategy.**
(DOCX)

**S2 Table. Serum vitamin E levels of different skin diseases.**
(DOCX)

**S3 Table. Quality of case control studies.**
(DOCX)

**S4 Table. Subgroup analysis of the association between serum vitamin E levels and skin diseases.**
(DOCX)

## Author Contributions

**Conceptualization:** Guang Yang.

**Data curation:** Xiaoxia Shi.

**Formal analysis:** Haoyuan Deng.

**Investigation:** Mengxin Luo, Qi Lan.

**Software:** Ningning Wang.

**Supervision:** Cong Zhang.

**Writing – original draft:** Xiaofang Liu.

**Writing – review & editing:** Xuezhu Xu, Cong Zhang.

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
