## [Decision Letter · Decision Letter 0]

31 Aug 2021

PONE-D-21-21726

Available information of serum vitamin E levels and chronic inflammatory dermatosis: systematic review and meta-analysis

PLOS ONE

Dear Dr. Zhang,

Thank you for submitting your manuscript to PLOS ONE. After careful consideration, we feel that it has merit but does not fully meet PLOS ONE’s publication criteria as it currently stands. Therefore, we invite you to submit a revised version of the manuscript that addresses the points raised during the review process.

Having intensively reviewed your revised draft, our external reviewers differed with their final recommendations, at least to some extent. Thus, I have double checked your revised version, to come to a more balanced decision (see R #1). All in all, our  identified shortcomings are considered reasonable with regard to both PLOS ONE’s quality standards and our readership's expectations. Therefore, we invite you to submit a carefully revised version of the manuscript that addresses EACH AND EVERY point raised during the current review process. Please note that a non-convincing revision (not considered acceptable with regard to language, content, reviewers' constructive criticisms, generalizable conclusions, and/or Authors' Guidelines) must lead to outright reject. 

We look forward to receiving your revised manuscript.

Kind regards,

Andrej M Kielbassa

Academic Editor

PLOS ONE

Journal Requirements:

2. Please consider modifying your title to ensure that it is specific, descriptive, concise, and comprehensible to readers outside the field (for example by clarifying the research question that this study aims to answer).

“The study was funded by Higher education reform project of 2018 (Liaoning, China)”

6. Please ensure that you refer to Figure 2, 3, 4 and 5 in your text as, if accepted, production will need this reference to link the reader to the figure.

7. We note you have included a table to which you do not refer in the text of your manuscript. Please ensure that you refer to Table 5 in your text; if accepted, production will need this reference to link the reader to the Table.

8. Please include your tables as part of your main manuscript and remove the individual files. Please note that supplementary tables (should remain/ be uploaded) as separate "supporting information" files.

Reviewers' comments:

Reviewer's Responses to Questions

**Comments to the Author**

1. Is the manuscript technically sound, and do the data support the conclusions?

Reviewer #1: No

Reviewer #2: Partly

Reviewer #3: Yes

2. Has the statistical analysis been performed appropriately and rigorously? 

Reviewer #1: Yes

Reviewer #2: Yes

Reviewer #3: Yes

3. Have the authors made all data underlying the findings in their manuscript fully available?

Reviewer #1: Yes

Reviewer #2: Yes

Reviewer #3: Yes

4. Is the manuscript presented in an intelligible fashion and written in standard English?

Reviewer #1: No

Reviewer #2: Yes

Reviewer #3: No

5. Review Comments to the Author

Reviewer #1: General remark

- English remains a concern, and several minor shortcomings regarding grammar and punctuation must be polished with a revised version. Remember that a flawless manuscript is the task of all co-authors, having read and approved the submission.

Abstract

- Please note the word maximum which is allowed with PLOS ONE (see Guidelines for Authors), to increase your information regarding your results, and to allow future readers to switch to your full text.

- Your aim was "To clarify the serum level of vitamin E in chronic inflammatory dermatosis, (...)." And you have concluded that "low serum Vitamin E levels increased the susceptibility risk of certain chronic inflammatory skin diseases." This would not seem convincing. Indeed, this would even seem meaningless. With your Conclusions, please stick exclusively to your aims. Do not simply repeat your results here. Do not provide banalities (known from each and every other paper). Instead, provide a reasonable and generalizable extension of your outcome.

Intro

- "Skin disease is one of the most common human illnesses." How can you say this? Please have a close look on caries, and on periodontitis. Again, please avoid common phrases seen with almost every paper.

- Please revise for referencing according to Journal style. "(...) pruritus, and so on[1, 2]." must read "(...) pruritus, and so on[1, 2]." First, make use of your spacebar, and revise thoroughly. Second, what do you mean when referring to "and so on"?

- Do not repeat statements. "(...) the relationship between serum vitamin E level and skin diseases is still unclear." and "However, the existing studies produced conflicting results." would provide comparable (or even the same) information. Please revise carefully, to facilitate reading.

- "So far, researches on level of serum vitamin E and skin diseases mainly focus on chronic inflammation skin diseases like vitiligo, psoriasis, atopic dermatitis and acne." References missing.

- Please clarify whether there already been any "systematic reviews of literature and meta-analyses" referring to vitamin E and dermal diseases. Make clear what you will add to the literature.

- Moreover, you surely had some idea prior to starting your study. What was your null hypothesis. Remember that H0 must be deducible from the foregoing thoughts. Revise carefully.

Meths

- Heading must read "Materials and methods". Again, you are strongly encouraged to consult the Journal's guidelines. Additionally, double checking some recently published PLOS ONE papers will help.

- "(...) by two authors (Qi Lan and Mengxin Luo)." must read "(...) by two authors (Q.L. and M.L.). Same with "C.Z.".

- "The literature search strategy was showed in Table. 1." must read "The literature search strategy is given with Table 1.". Revise carefully, to avoid any typos/delete full stops not considered necessary.

- "(version 13, StataCorp LP, College Station, TX)." must read "(Stata Statistical Software, Release 13; StataCorp LP, College Station, TX, USA)."

Results

- "In quality assessment, scores of all included case-control studies are showed in Table 3" Please clarify the quality grades with your text. Remember that only high-quality papers should be included, to avoid repetition of any "conflicting results" (which have been deemed by the Authors). Please clarify, and discuss.

- "(P < 0.00)" would seem unclear. As a general rule, please always provide p values on a 3-digit basis. Format must be "(p = 0.838)", or "(p < 0.001)". Note that lower case "p" would fit to the Journal style.

- High heterogeneity would seem a problem, and there must be a convincing rationale to proceed with your paper. Please discuss.

- "As a sensitivity analysis, a meta-influence plot was used to analyze the influence of individual studies on the overall effect size (Figure. 7)." This would not seem acceptable. Please remember that it is considered the task of the Authors to guide the reader. Simply referring to a Table, or to a Figure (without providing any explanations) is not deemed professional.

Disc

- Refer to H0 with the first paragraph of this section.

- "It was found that >60% of adults have vitamin E intakes below the EAR (<12 mg/d) in the United States." Again, each and every statement calls for reference(s). Revise thoroughly throughout your text.

- "In our study, the lower serum vitamin E levels in patients of vitiligo, psoriasis, atopic dermatitis and acne compared with controls." Please double check and revise sentence.

- This section has not been thoroughly elaborated, and there would seem room for more profound discursive aspects.

Concl

- Please see comments given above. Again, do not simply repeat your results here - these already have been presented (and should be thoroughly discussed, see comments given above). Instead, provide a reasonable and generalizable extension of your outcome.

- Same with "It might be due to the number of researches on (...)." Explanations (and even speculations) are welcome, but must be provided with the Disc section.

- Same with "The low quality and high heterogeneity of some included studies means that our results must be interpreted with caution (...)." Again, this is not considered a Conclusion. You should discuss these aspects at the right place. Why did you include those poor papers? Why did you finish such a study involving several poor papers? Please provide a sound and reasonable rationale. Moreover (and again), this is a clear limitation, and the Authors must thoroughly clarify, why PLOS ONE should proceed with this draft. At the end of the day, this only will be another paper considered "unclear", "controversial", or "conflicting". Please see comments given above.

- "More studies of high-quality observational are required to confirm the association between low serum vitamin E levels and skin diseases." Again, this is a meaningless platitude. And, moreover, this would clarify that your study did not reveal any associations, right?

Refs

- Please revise for uniform formatting. Consult some previously published PLOS ONE papers.

Reviewer #2: This systematic review focuses on the relation between vitamin E levels and chronic inflammatory dermatoses.

It is widely recognised that it would be better to study dietary patterns rather than single nutrients since nutrients and foods may interact in their biological effects.

Reverse causation is a major issue and should be clearly discussed in the study. Reverse causation can occur when people change their diet or other lifestyle habit after developing a disease or perhaps after having a close family member suffer an event like vitiligo or other immune-related conditions

Reviewer #3: This is a commendable review by the authors to understand how vitamin E can relate to various skin issues. There are a couple revisions that are needed before this article can be considered for acceptance:

1) The English is poor. I can't detail every single grammatical error as they are numerous. Please have a native English speaker review this for grammatical accuracy.

2) The authors should not refer to vitamin E as if it is just one molecule. Please detail the subsets of vitamin E to detail what was actually measured. Was is alpha tocopherol, gamma tocopherol, tocotrienol subvariants, etc?

6. PLOS authors have the option to publish the peer review history of their article (what does this mean?). If published, this will include your full peer review and any attached files.

Reviewer #1: No

Reviewer #2: No

Reviewer #3: No

---

## [Author Response · Author response to Decision Letter 0]

24 Oct 2021

We uploaded our respond to reviewers as an attachment.

---

## [Decision Letter · Decision Letter 1]

10 Nov 2021

PONE-D-21-21726R1Serum vitamin E levels and chronic inflammatory skin diseases: A systematic review and meta-analysis

PLOS ONE

Dear Dr. Zhang,

Thank you for submitting your manuscript to PLOS ONE. After careful consideration, we feel that it has merit but does not fully meet PLOS ONE’s publication criteria as it currently stands. Therefore, we invite you to submit a revised version of the manuscript that addresses the points raised during the review process.

Having intensively reviewed your revised draft, our external reviewers basically have agreed with their final recommendations. Additionally, I have double checked your revised version, to come to a final decision (see R #1). All in all, I am still convinced that your revised paper will be worth following, even if your revised version still would benefit from minor re-edits and some polishing.

We look forward to receiving your revised manuscript.

Kind regards,

Andrej M Kielbassa

Academic Editor

PLOS ONE

Journal Requirements:

Reviewers' comments:

Reviewer's Responses to Questions

**Comments to the Author**

1. If the authors have adequately addressed your comments raised in a previous round of review and you feel that this manuscript is now acceptable for publication, you may indicate that here to bypass the “Comments to the Author” section, enter your conflict of interest statement in the “Confidential to Editor” section, and submit your "Accept" recommendation.

Reviewer #1: All comments have been addressed

Reviewer #3: All comments have been addressed

Reviewer #4: All comments have been addressed

2. Is the manuscript technically sound, and do the data support the conclusions?

Reviewer #1: Yes

Reviewer #3: Yes

Reviewer #4: Yes

3. Has the statistical analysis been performed appropriately and rigorously? 

Reviewer #1: Yes

Reviewer #3: Yes

Reviewer #4: Yes

4. Have the authors made all data underlying the findings in their manuscript fully available?

Reviewer #1: Yes

Reviewer #3: Yes

Reviewer #4: Yes

5. Is the manuscript presented in an intelligible fashion and written in standard English?

Reviewer #1: Yes

Reviewer #3: Yes

Reviewer #4: Yes

6. Review Comments to the Author

Reviewer #1: With the help of the reviewers, this revised and re-sublmitted draft has been considerably improved, and would seem ready to be forwarded to the external referees.

Reviewer #3: Thank you for the edits and this is suitable for publication. The authors now delineate the different subsets of vitamin E more clearly and the English is improved.

Reviewer #4: In this study, a comprehensive systematic review and metaanalysis was conducted to evaluate the effects of vitamin e levels on chronic inflammatory skin diseases in human subjects. This review is well written. Also, in the first revise, the reviewer's comments were responded appropriately. To the reviewer's knowledge, there has been no systematic review on this topic in the past, and it will provide important knowledge and useful information for this research area. However, there are some comments that could elevate the quality of this paper. Reviewer added recommendations and suggestions regarding this.

Comment 1

This review focuses primarily on chronic inflammatory skin diseases. Are there any reported synergies between chronic inflammatory skin diseases and other physiological effects (such as lifestyle disease and immunologic disease)? It would be more interesting for the reader to add these explanations in the first half of the article.

Comment 2

Reviewer concern about the subjects’ characteristics used for this analysis. Authors mentioned about their age but less information about their sex even though some investigations the number of male/ female is considerably biased. Was the sex a considerable factor?

Comment 3

Reviewer felt the resolution of each figure was low. Reviewer recommends that the author replace it with a higher resolution figure.

Comment 4

The authors state that the effects of vitamin E on chronic inflammatory skin diseases are predominantly antioxidant and anti-inflammatory.

On the other hand, for example, in the following review reported this year,

Reflections on a century of vitamin E research: Looking at the past with an eye on the future,

Free Radical Biology and Medicine,

Volume 175, 2021, Pages 155-160, ISSN 0891-5849,

doi.org/10.1016/j.freeradbiomed.2021.07.042.

https://www.sciencedirect.com/science/article/pii/S0891584921007048?via%3Dihub

These review mention that vitamin E's antioxidant and anti-inflammatory effects do not work in the body. What do the authors think of such a debate?

7. PLOS authors have the option to publish the peer review history of their article (what does this mean?). If published, this will include your full peer review and any attached files.

Reviewer #1: No

Reviewer #3: No

Reviewer #4: No

---

## [Author Response · Author response to Decision Letter 1]

24 Nov 2021

The respond to reviewers has been uploaded.

---

## [Decision Letter · Decision Letter 2]

26 Nov 2021

Serum vitamin E levels and chronic inflammatory skin diseases: A systematic review and meta-analysis

PONE-D-21-21726R2

Dear Dr. Zhang,

We’re pleased to inform you that your manuscript has been judged scientifically suitable for publication and will be formally accepted for publication once it meets all outstanding technical requirements.

Kind regards, congratulations, and stay healthy

Prof. Dr. med. dent. Dr. h. c. Andrej M. Kielbassa

Academic Editor

PLOS ONE

Reviewers' comments:

Reviewer's Responses to Questions

**Comments to the Author**

1. If the authors have adequately addressed your comments raised in a previous round of review and you feel that this manuscript is now acceptable for publication, you may indicate that here to bypass the “Comments to the Author” section, enter your conflict of interest statement in the “Confidential to Editor” section, and submit your "Accept" recommendation.

Reviewer #1: All comments have been addressed

Reviewer #4: All comments have been addressed

2. Is the manuscript technically sound, and do the data support the conclusions?

Reviewer #1: Yes

Reviewer #4: Yes

3. Has the statistical analysis been performed appropriately and rigorously? 

Reviewer #1: Yes

Reviewer #4: Yes

4. Have the authors made all data underlying the findings in their manuscript fully available?

Reviewer #1: Yes

Reviewer #4: Yes

5. Is the manuscript presented in an intelligible fashion and written in standard English?

Reviewer #1: Yes

Reviewer #4: Yes

6. Review Comments to the Author

Reviewer #1: The Authors have re-submitted a revised draft which has been considerably improved, not only with the help of the reviewers. This manuscript is ready to proceed.

Reviewer #4: Reviewer confirmed the authors' reply and the revised manuscript in detail. The authors have responded and revised appropriately to the reviewers' comments. Thank you for the revision.

7. PLOS authors have the option to publish the peer review history of their article (what does this mean?). If published, this will include your full peer review and any attached files.

Reviewer #1: No

Reviewer #4: No

---

## [Editor Report · Acceptance letter]

2 Dec 2021

PONE-D-21-21726R2 

Serum vitamin E levels and chronic inflammatory skin diseases: A systematic review and meta-analysis 

Dear Dr. Zhang:

I'm pleased to inform you that your manuscript has been deemed suitable for publication in PLOS ONE. Congratulations! Your manuscript is now with our production department. 

Kind regards, 

on behalf of

Prof. Dr. med. dent. Dr. h. c. Andrej M Kielbassa 

Academic Editor

PLOS ONE